**communications** engineering

# Automatic design of stigmergy-based behaviours for robot swarms
Muhammad Salman [1,2,3], David Garzón Ramos [1,3] & Mauro Birattari [1] ✉

Stigmergy is a form of indirect communication and coordination in which individuals influence their peers by modifying the environment in various ways, including rearranging objects in space and releasing chemicals. For example, some ant species lay pheromone trails to efficiently navigate between food sources and nests. Besides being used by social animals, stigmergy has also inspired the development of algorithms for combinatorial optimisation and multi-robot systems. In swarm robotics, collective behaviours based on stigmergy have always been designed manually, which is time consuming, costly, hardly repeatable, and depends on the expertise of the designer. Here, we show that stigmergy-based behaviours can be produced via automatic design: an optimisation process based on simulations generates collective behaviours for a group of robots that can lay and sense artificial pheromones. The results of our experiments indicate that the collective behaviours designed automatically are as good as—and in some cases better than—those produced manually. By taking advantage of pheromone-based stigmergy, the automatic design process generated collective behaviours that exhibit spatial organisation, memory, and communication.

Stigmergy[1–3] is a coordination mechanism in which agents self-organise through indirect local communication mediated by the environment. When using stigmergy, agents leave indications of their presence or actions in the environment and this stimulates/inhibits the behaviours of their peers[4]. Some animals physically transform the environment thus producing visual cues that influence their peers. For instance, humans leave footprints on the ground and flatten vegetation while walking in the wild, thereby creating visually detectable paths that others tend to follow[5]. Other animals secrete chemicals that their peers can detect and to which they react—for instance, Argentine ants lay pheromone trails that are then followed by nestmates[6].

For many social insects, pheromone-based stigmergy plays an important role in self-organisation[7]. These insects can sense environmental features, locally interact with other members of the colony and with the environment, and process information to make decisions[8]. However, they have short perception and communication ranges, are not aware of the global state of the colony, are unable to remember their actions, and are unable to plan their contributions to the collective activities of the colony[8]. The pheromones laid in the environment function as a collective and distributed memory: they effectively encode the state of the colony. The pheromones enable coordination, as the individuals can work together and self-organise without the need to communicate directly or receive instruction on the tasks they must perform[9,10].

In a robot swarm, which operates similarly to a colony of social insects[11], a collective behaviour emerges due to local interactions between individual robots and between the robots and the environment[12]. A robot swarm, like an insect colony, can use pheromone-based indirect communication mediated by the environment[13]. Designers of robot swarms can develop pheromone-based interaction strategies for specific missions. However, giving real robots the capability to mark the environment with indications of their activities is still an open technological challenge[14]. In some studies, researchers have developed smart environments to enable pheromone-based stigmergy, for instance, by using: (i) a system of stationary devices (e.g., RFID tags) spread throughout the environment to store virtual pheromones[15–19], (ii) devices to display or project virtual pheromones on the ground[20–23], or (iii) augmented reality to immerse the robots in a virtual environment in which they can lay and sense pheromones[24–26]. These systems are flexible, powerful, and enable the implementation of complex coordination mechanisms. However, as these systems rely on external infrastructures (for tracking robots, displaying the pheromones, and storing information), they can be expensive and are only suitable under restricted conditions. Alternatively, several approaches to physically deposit artificial pheromones have been proposed, using specialised onboard actuators to lay trails of alcohol or wax, without the assistance of any external infrastructure[27–29]. However, these solutions would be impractical in most real-world applications due to the hazards of using flammable material

[1]IRIDIA, Université libre de Bruxelles (ULB), Brussels, Belgium. [2]Institute of Astronomy, KU Leuven, Leuven, Belgium. [3]These authors contributed equally: Muhammad Salman, David Garzón Ramos. ✉e-mail: Mauro.Birattari@ulb.be

(alcohol) or heating devices (for melting wax). To address the issue, we have recently proposed a hardware module for robots that project UV light downwards, laying an artificial pheromone trail on ground that has previously been coated with photochromic material[30]. The part of the ground that is exposed to UV light changes in colour from white to magenta. Once the UV light is removed, the magenta colour fades back to white, in about 50 s, mimicking the evaporation of a pheromone. This approach does not present safety risks and does not rely on complex or expensive infrastructure, however, it still requires the environment to be prepared before deploying the robots.

The technological problem of endowing the robots with the ability to lay and sense artificial pheromones is not the only problem to be addressed. The concept of stigmergy is not easily understood intuitively[31] and therefore, designing collective behaviours based on stigmergy is itself a challenge. Even without using stigmergy, designing any collective behaviour for a robot swarm is already complex: individuals are autonomous and loosely coupled, and the interactions between individuals and between them and the environment become fully defined only at run time[32,33]. The design problem becomes even more complex if the interaction strategies that enable coordination are regulated by modifications to the environment. No formal design method exists to tell under what conditions and in what amount individuals should release the pheromone, nor how they should react to pheromone trails so that a desired collective behaviour emerges. In the swarm robotics literature, pheromone-based stigmergy has been predominantly designed manually, via trial and error, to address specific missions under specific conditions[26,34,35]. Manual design is a time-consuming approach in which a human designer conceives, tests, and iteratively improves the control software of the robots, until a desired collective behaviour is obtained[36,37]. The quality of the results obtained via manual design is not consistent and greatly depends on the experience of the designer. Typically, a manual design process is neither easily repeatable, nor directly generalisable to other—albeit similar—robotic platforms or missions[38]. The only exception to manual design is one study in which deep reinforcement learning was used to develop a collision avoidance behaviour based on a virtual pheromone[39]. Although restricted to simulation-only experiments, this study showed that control software produced through deep reinforcement learning can outperform the one generated via manual design. The proposed approach was conceived for scenarios where a centralised infrastructure stores global pheromone information and makes it accessible to the robots. On the one hand, this approach provides a solution to the problem of designing pheromone-based behaviours in virtual environments. On the other hand, the approach is not directly applicable in scenarios where the robots are expected to autonomously lay and sense the artificial pheromones in their physical environment.

In this paper, we focus on the automatic design of stigmergy-based collective behaviours for robot swarms. We present `Habanero`, an automatic off-line design method that belongs to the AutoMoDe family[40]. In AutoMoDe, as is customary in automatic off-line design[38,41], the design problem is reformulated as an optimisation problem that is solved in simulation, prior to the deployment of the robots in their target environment[41,42]. The solution space of the optimisation problem comprises instances of control software that can be obtained by selecting and combining pre-existing software modules (i.e., low-level behaviours and the conditions to transition between them) into a modular architecture (e.g., finite-state machines, behaviour trees) and by tuning the free parameters[43]. Once the optimisation process is completed, the selected control software is uploaded to the robots without undergoing any manual transformations, and the robots are eventually deployed in the target environment. It has been observed that the control software produced by AutoMoDe crosses the reality gap[44–47] better than traditional approaches based on neuroevolution[43,48], in which robots are controlled by a neural network that is optimised using an evolutionary algorithm[49–51]. This improvement can be attributed to AutoMoDe's constraint that control software must be generated by assembling the given modules within a specific architecture (e.g., a probabilistic finite-state machine). By applying this constraint, AutoMoDe limits the size of the design space to the set of possible combinations of modules, and therefore reduces the variance of the design process[43]. This reduces the risk of over-fitting the control software produced to the idiosyncrasies of the simulation environment, which is the main reason why control software might fail to cross the reality gap satisfactorily[47].

AutoMoDe is a general framework. To define a specific design method that conforms to it and produces control software to address a specific class of missions, the following steps must be taken: (1) select a target robot platform that is appropriate for the given class of missions, (2) define software modules for the selected robot platform, (3) specify the architecture into which the software modules will be assembled, (4) select a simulator to be used in the automatic design process, and (5) define an appropriate optimisation algorithm to search the space of the possible ways in which the software modules can be assembled and tuned. Our proposed AutoMoDe method, `Habanero`, designs collective behaviours to address missions in which the robot swarm relies on stigmergy to coordinate. The target robot platform is the e-puck[52] augmented with the Overo Gumstix Linux board, the aforementioned hardware module that lays artificial pheromone trails by focusing UV light onto ground coated with photochromic material[30], and an omnidirectional camera to detect artificial pheromone trails. The software modules of `Habanero` are based on those previously defined for `TuttiFrutti`[53], another AutoMoDe method that generates control software for robots that can display colours via RGB LEDs and react to them. The main difference between `TuttiFrutti` and `Habanero` is that the latter features some original hardware and software devices to lay and detect pheromone trails. The architecture into which these modules are assembled are probabilistic finite-state machines. The simulator used in the design process is ARGoS[54] with an original library for the simulation of pheromone trails. The optimisation algorithm utilised is Iterated F-race[55], as originally used in `TuttiFrutti`[53] and in `Chocolate`, the state-of-the-art AutoMoDe method[56]. See Fig. 1 for a graphical illustration of `Habanero`, Fig. 2 for a description of the platform for which `Habanero` was developed, and the Methods section for further details. The collective behaviours designed by `Habanero` enable the robots to operate in a fully autonomous and distributed way without requiring any form of centralised control and coordination.

In this study, we demonstrate `Habanero` by generating control software for a swarm of eight e-puck robots. We consider four missions in which the robots should rely on stigmergy-based coordination: AGGREGATION, DECISION MAKING, RENDEZVOUS POINT, and STOP. See Fig. 3 and the Methods section for details. To assess the quality of the control software produced by `Habanero`, we compare its performance to that of several alternatives, shown in Fig. 4: (1) control software produced via neuroevolution (`EvoPheromone`), (2) control software manually produced by human designers (`Human-Designers`), and (3) a random-walk behaviour (`Random-Walk`).

The results of the experiments indicate that: (i) `Habanero` is a viable approach to designing pheromone-based stigmergy; (ii) it can produce control software that is comparable to, or even outperforms, control software produced by a human designer; and (iii) although its modules are conceived in a mission-agnostic way, the interaction strategies it devises are mission-specific.

## Results

`Habanero` designed stigmergy-based collective behaviours that proved to be effective: the robots used the artificial pheromone to complete each mission in a way that is meaningful and appropriate to the mission considered. Statistical analysis shows that the control software generated by `Habanero` performed significantly better than the alternatives included in the empirical study. In the following sections, we first present the results on a per-mission basis, and then we aggregate them across all missions. Simulation-only experiments with different swarm sizes are provided as Supplementary Note 1. We also provide an analysis of the robustness to the reality gap as Supplementary Note 2.

## AGGREGATION

In this mission, the robots must aggregate anywhere in the arena. To aggregate, the robots cannot rely on any form of direct communication nor on the ability to directly sense the presence of their peers in their vicinity. The only way in which they can coordinate is the laying and detecting of artificial pheromone trails. They can leverage this ability to attract their peers and aggregate using stigmergy. However, as all robots could release some pheromone at the same time in different areas, they could saturate the environment and/or be trapped in the local accumulation of their own pheromone emissions.

`Habanero`, `EvoPheromone`, and `Human-Designers` produced control software that performed equivalently well in simulation—see Fig. 5a. However, when transferred to the real robots, the control software produced by `Habanero` performed significantly better than the one produced by all other design methods.

`Habanero` produced collective behaviours in which the robots laid pheromone trails only for short periods of time and kept searching the environment for pheromone traces left by their peers. By laying pheromone trails only intermittently, the robots avoided saturating the environment and marked only isolated spots, which then served as aggregation points. Around these points, they eventually gathered in clusters—see Fig. 6 and Supplementary Video 1.

`EvoPheromone` produced a different strategy: the robots laid pheromone trails while moving along a circular trajectory and followed the pheromone trails to gather at places where pheromone concentration was high. This strategy produced good results in simulation but not on the real robots. The robots did not properly avoid the walls and failed to reproduce the behaviour observed in the simulation.

The control software produced by `Human-Designers` continuously laid pheromone trails with the expectation that all robots would gather at one place. Results were good in simulation but failed to transfer to reality. In the real-robot experiments, the robots remained trapped in local pheromone accumulations. Eventually, they gathered in separate clusters.

## DECISION MAKING

In this mission, the robots must make the decision to congregate in one of two regions of the arena, designated by RGB blocks that display blue or green colour, respectively—see Fig. 3b. Each robot scores one point for each time step spent in the green region and two points for each time step spent in the blue one. Halfway through each run of the experiment, the blue and green RGB blocks are switched off, leaving the robots without any visual cue to identify the two regions. In order to maximise the score, the robots must quickly congregate in the region that provides the highest score per time step

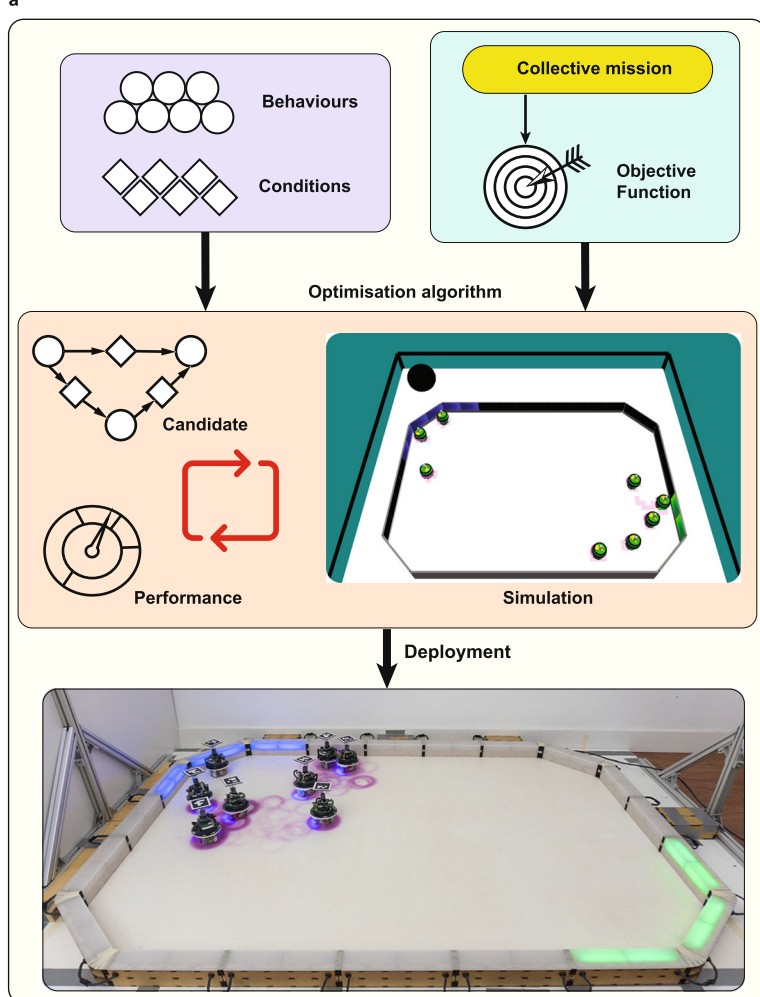

**b**

| Behaviours | | |
|---|---|---|
| Exploration | EXP | Robot moves by random walk |
| Stop | STP | Robot stops on the spot |
| Go-to-Colour | GTC | Robot moves toward a specific colour |
| Avoid-Colour | AVC | Robot moves away from a specific colour |
| Go-to-Pheromone | GTP | Robot moves toward pheromone |
| Avoid-Pheromone | AVP | Robot moves away from pheromone |
| Waggle | WGL | Robot rotates in place for a random period |

**c**

| Transition Conditions | | |
|---|---|---|
| White-Floor | WF | White floor detected |
| Gray-Floor | GF | Gray floor detected |
| Black-Floor | BF | Black floor detected |
| Colour-Detected | CD | Objects of a specific colour perceived |
| Pheromone-Detected | PD | Pheromone perceived |
| Fixed-Probability | FP | Transition with a fixed probability |

**Fig. 1 | AutoMoDe-Habanero. a** `Habanero` automatically produces control software for e-puck robots by assembling predefined and mission-independent software modules into a probabilistic finite-state machine. A set of seven low-level behaviours and six transition conditions function as states and edges of the finite-state machine, respectively. Using the Iterated F-race algorithm, the design process determines the topology of the finite-state machine by maximising the performance of the robot swarm. The performance of an instance of control software is assessed in simulation, before the swarm is deployed. **b** The set of low-level behaviours: operations a robot can execute. **c** The set of transition conditions: criteria to switch from a low-level behaviour to another one.

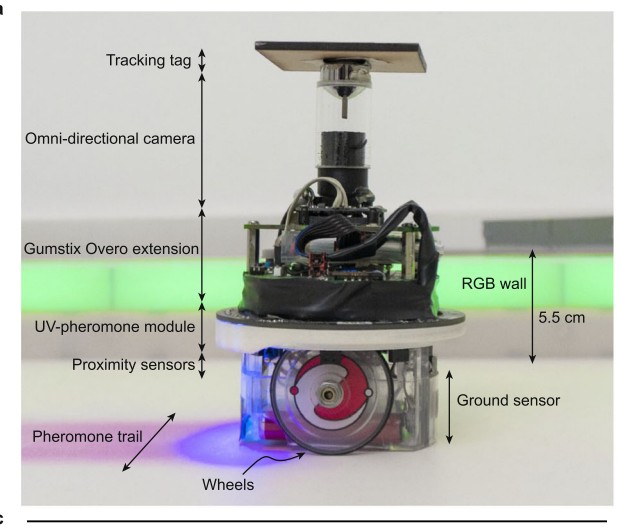

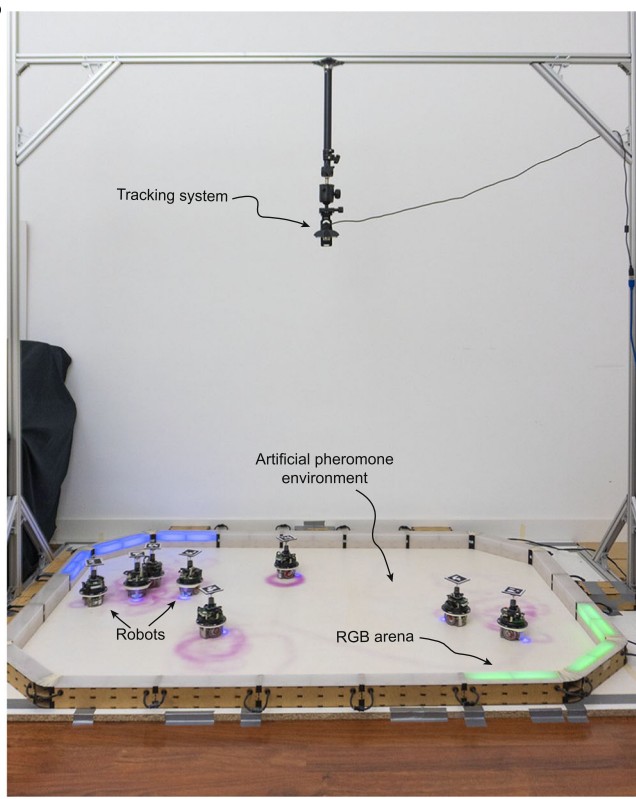

| Sensor | Input | Value |
|---|---|---|
| Proximity | $prox_{i \in \{1,\ldots,8\}}$ | $[0, 1]$ |
| Ground | $ground_{j \in \{1,2,3\}}$ | $\{black, gray, white\}$ |
| Omni-directional camera | $cam_{c \in \{R,G,B,C,M,Y\}}$ | $\{yes, no\}$ |
| | $V_{c \in \{R,G,B,C,M,Y\}}$ | $(1.0; [0, 2]\,\pi\,\mathrm{rad})$ |

| Actuator | Output | Value |
|---|---|---|
| Motors | $v_{k \in \{l,r\}}$ | $[-0.12, 0.12]\,\mathrm{m\,s}^{-1}$ |
| UV-Pheromone-Module | $phe$ | $\{none, thin, thick\}$ |

**Fig. 2 | The e-puck robot, its reference model, and the experimental setup. a** An e-puck robot equipped with a Linux board, a hardware module to focus UV light onto the ground, and an omni-directional camera. **b** The experimental arena. The floor is coated with photochromic material. It changes in colour from white to magenta when exposed to UV light, and gradually returns to its normal white colour when the UV light is removed. The walls of the arena are constructed using modular RGB (Red, Green, Blue) blocks, which have the ability to display various colours using the RGB colour code. A tracking system is used to automatically measure performance indicators. **c** The reference model RM 4.1, which formally describes the interface between the robot and the control software.

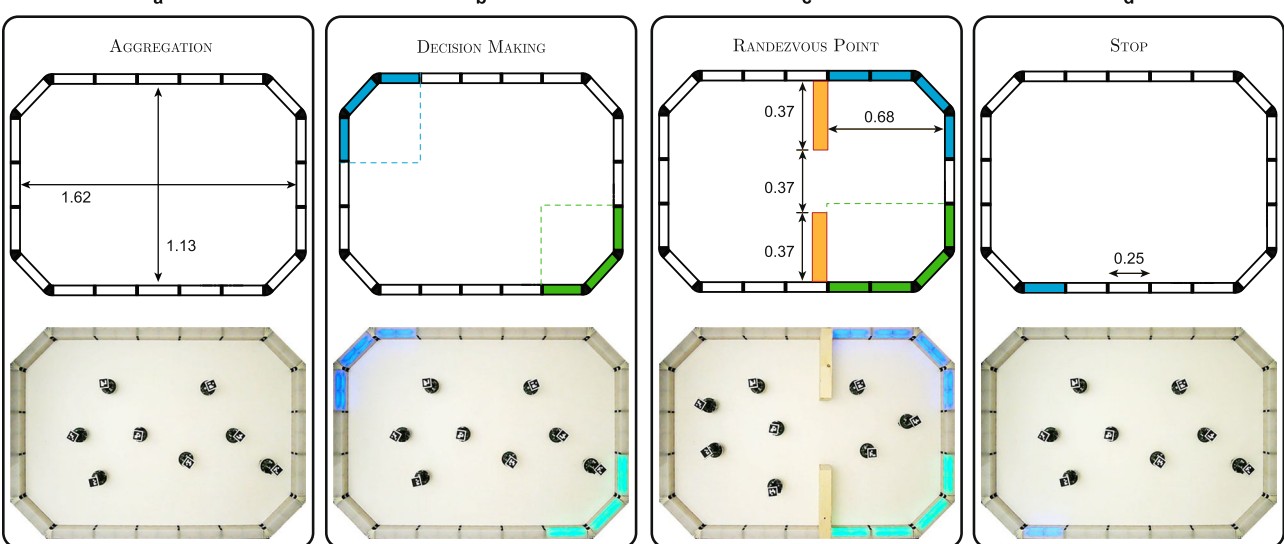

**Fig. 3 | Construction of the arenas for the four missions.** Technical drawings of the arena with dimensions and positions of different regions, along with photos of the real arena, in the four mission configurations: **a** AGGREGATION, **b** DECISION MAKING, **c** RENDEZVOUS POINT, and **d** STOP. All measurements are expressed in meters. The missions are described in the Methods section.

—i.e., the blue one—and remain there even once the environmental cues are removed.

When evaluated in simulation, the control software produced by `Habanero` and `Human-Designers` performed equally well, and significantly better than the one produced by `EvoPheromone`—see Fig. 5b. However, in the real-robot experiments, the control software produced by `Habanero` performed significantly better than that of `Human-Designers`. The control software produced by both `Habanero` and

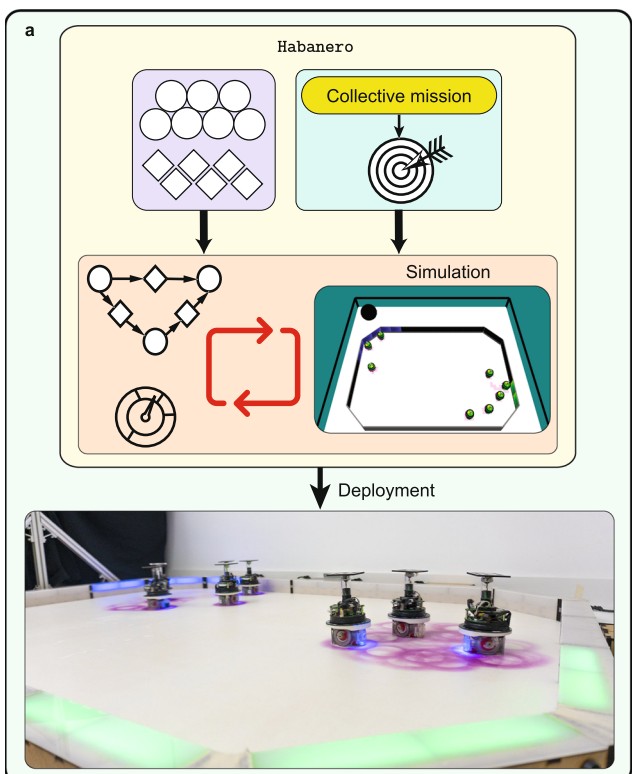

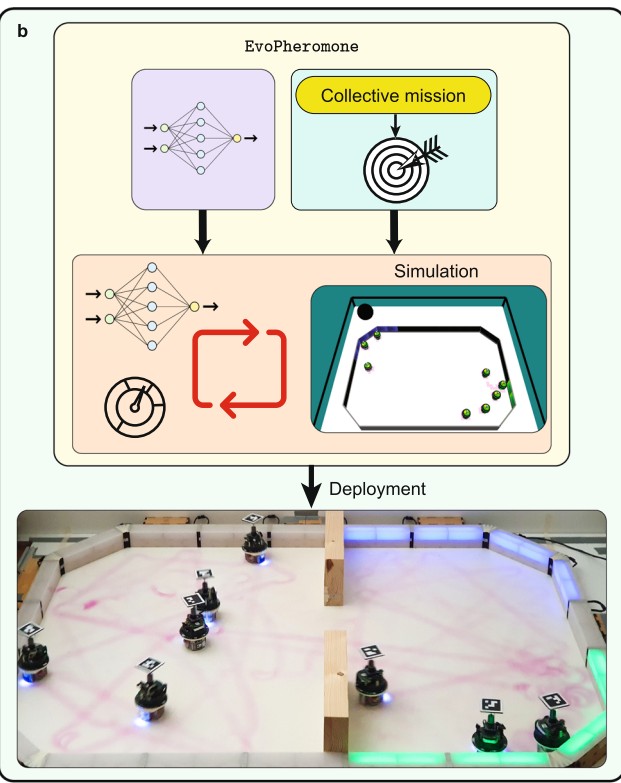

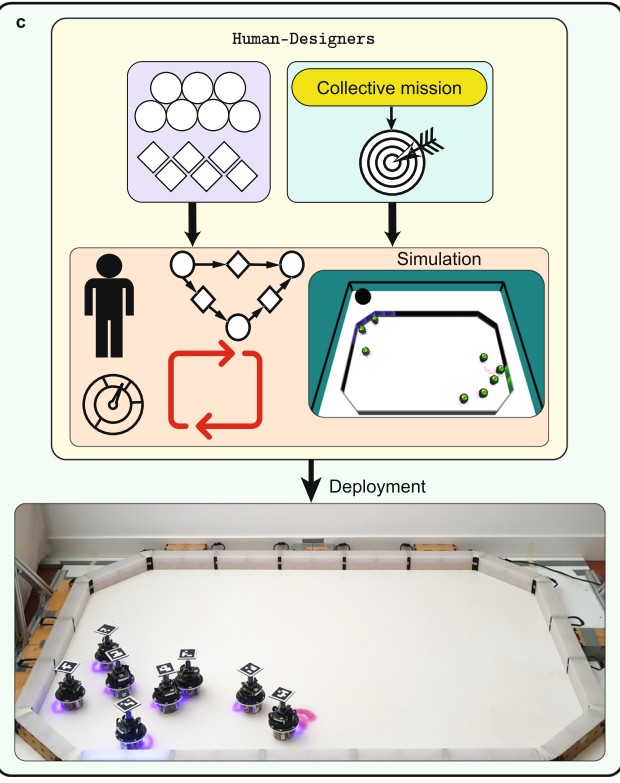

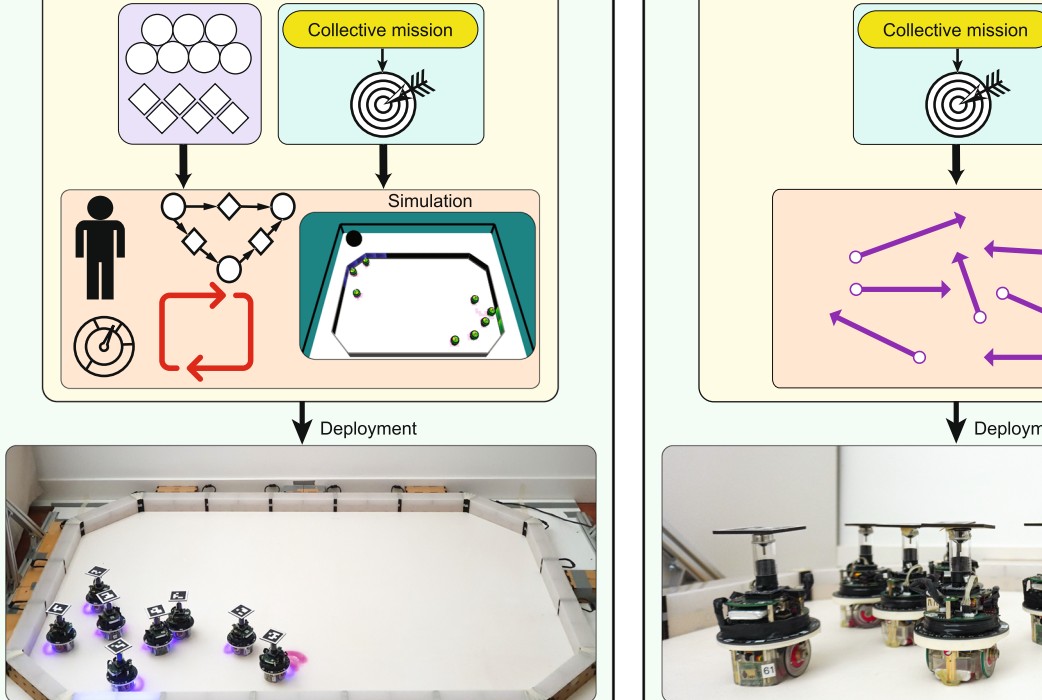

**Fig. 4 | Pictorial representation of the design methods under analysis.**
**a** `Habanero`, **b** `EvoPheromone`, **c** `Human-Designers`, and **d** `Random-Walk`.
`Habanero` is an automatic modular design method. `EvoPheromone` is an
implementation of the neuroevolutionary approach. `Human-Designers` is a
manual design method. Although `Random-Walk` is not a design method, we include
it to serve as a lower bound of performance. See the Methods section for the details.

`Human-Designers` performed significantly better than that of `Evo-Pheromone`, which obtained results comparable with those of `Random-Walk`.

In all experimental runs, the robot swarm designed by `Habanero` correctly selected the blue region to congregate. The robots relied on

stigmergy not only to attract other robots to the blue region, but also to stay there after the cues were removed. The behaviour displayed in the real-robot experiments was qualitatively similar to the one displayed in simulation—see Fig. 6 and Supplementary Video 2. However, in the real-robot experiments, some robots that gathered in the blue region spilled out of the

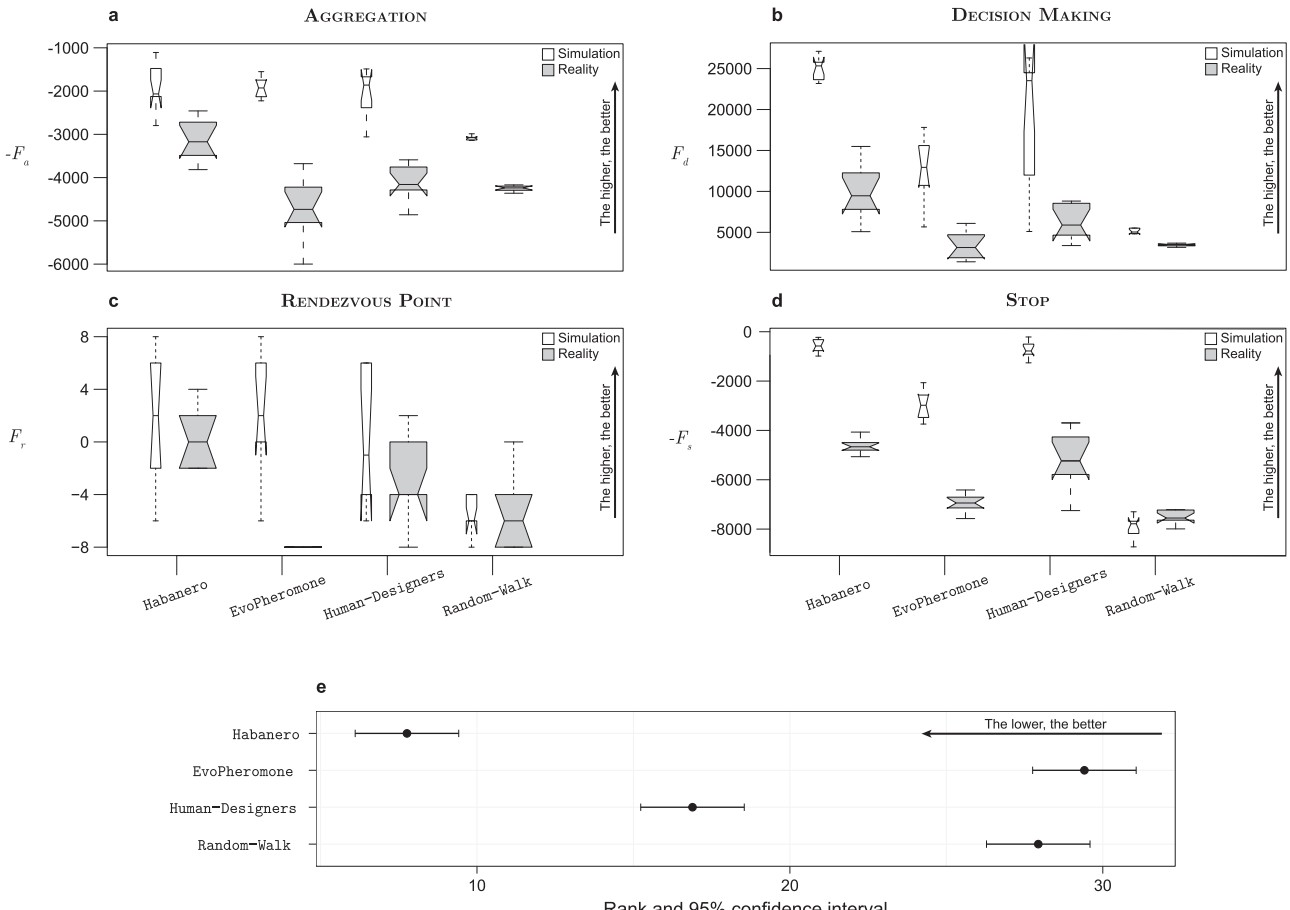

**Fig. 5 | Results of the empirical analysis.** We report results of the evaluation of 160 instances of control software, 10 per method and per mission. All instances of control software were evaluated once in simulation and once with physical robots— more details on the protocol are provided in Methods. The results are presented using boxplots on a per-mission basis: **a** AGGREGATION, **b** DECISION MAKING, **c** RENDEZVOUS POINT, and **d** STOP. In all missions, for each method, we report the performance obtained in simulation and with physical robots using thin and thick boxes, respectively. **e** Friedman rank sum test on real-robot performance to aggregate the overall performance of each method across the four missions—the lower the rank, the better. An explanation of the graphical convention adopted in the boxplots and in the Friedman test are provided in the Methods section under the heading Statistics.

boundaries of the region, although remaining in its vicinity. Because of this, the performance in the real-robot experiments was lower than that in simulation. The robot swarm generated by Habanero was unable to congregate in a single region: the robots stayed in the first region in which they entered. Consequently, the score was significantly worse than the one obtained by other design methods. The robot swarm produced by Human-Designers was able to correctly congregate in the blue region but was unable to remain there once the cues were removed.

### RENDEZVOUS POINT

In this mission, a wall with a narrow gate laterally divides the arena into two sections: the left side, where the robots are deployed at the beginning of the experiment; and the right side, which contains two regions designated by RGB blocks that display blue or green colour, respectively—see Fig. 3c. Similar to DECISION MAKING, halfway through each run of RENDEZVOUS POINT, the blue and green RGB blocks are switched off, leaving the robots without any visual cue to identify the two regions. The robots must cross the narrow gate to gather in the green region. The score is given by the number of robots that, at the end of the experimental run, are positioned in the green region.

When evaluated in simulation, the control software produced by all design methods performed equally well—see Fig. 5c. However, in the real-robot experiments, the control software produced by Habanero

performed significantly better than the one produced by all other methods. Moreover, the one produced by EvoPheromone performed significantly worse than that produced by all other methods.

The robot swarms designed by Habanero relied on random walk to cross the gate and find the green region. Once the robots reached the green region, they took advantage of stigmergy to attract their peers and to keep themselves inside the region even when the green light was removed. The robots laid pheromone trails to mark the green region and kept laying the pheromone trails at that place to avoid fading—see Fig. 6 and Supplementary Video 3.

In the control software produced by EvoPheromone, the robots do not randomly search for the narrow passage. Instead, they move along the walls of the arena to eventually cross the gate and reach the green region— see Supplementary Video 3. Although this behaviour worked effectively in simulation, it failed in the real-robot experiments: the robots were unable to move along the walls and remained stuck. Consequently, they were unable to cross the gate. In the real-robot experiments, the performance of the robot swarm designed by EvoPheromone was even significantly worse than that of Random-Walk.

In the control software produced by Human-Designers, the robots were mostly able to reach the green region. However, the swarm produced by Human-Designers was not always effective in using stigmergy to remain in the green region, especially after the green light was removed.

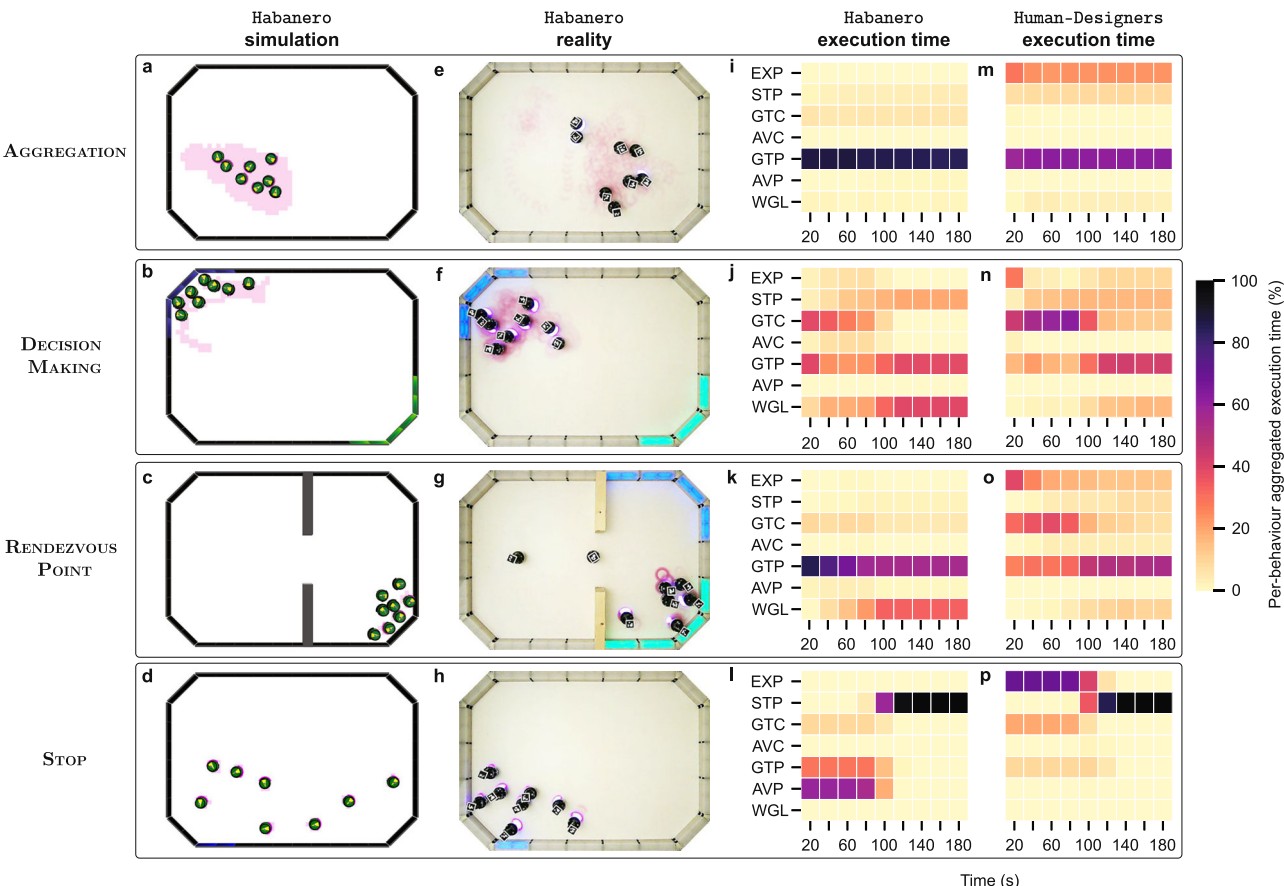

**Fig. 6 | Behaviours produced by** `Habanero` **and** `Human-Designers`. For each mission, we show: a snapshot of robots executing an instance of `Habanero` control software in (**a–d**) simulation and (**e–h**) real-robot experiments, as well as a plot of the aggregate execution time of each software module in the control software produced by (**i–l**) `Habanero` and (**m–p**) `Human-Designers`. We use the aggregate execution time of the modules to qualify the behaviour we observe in the robot swarms. In the aggregate plots, the colour gradient shows the percentage of time one behaviour was executed throughout all instances of control software produced for a mission. We identify the behaviour modules using the labels defined in Fig. 1.

## STOP

In this mission, the robots must halt and stand still as soon as a stop signal is perceived. The stop signal is a (random) RGB block that switches on at a random moment in time and emits blue light—see Fig. 3d. Before the signal, each robot scores one point for each time step during which it moves. After the signal, each robot scores one point for each time step during which it stays in place. As the robots considered in this study are incapable of direct communication, the individuals that detect the signal can only rely on stigmergy to inform any peers that are in a position from which the signal cannot be seen.

The control software produced by `Habanero` and `Human-Designers` performed similarly well when evaluated both in simulation and reality, and performed significantly better than the one produced by `EvoPheromone`—see Fig. 5d.

In the robot swarms designed by `Habanero`, the robots kept moving to search for a block emitting the stop signal. As soon as a robot detected the signal, it stopped or started waggling in place, while laying a pheromone trail to alert its peers. Other robots also stopped and started laying pheromone trails either after detecting the signal or the pheromone trails laid by their peers—see Supplementary Video 4.

`Human-Designers` produced collective behaviours similar to those generated by `Habanero`, and so no significant difference in the performance could be observed—see Fig. 5d.

The collective behaviours produced by `EvoPheromone` achieved good scores in some cases, but were unable to accomplish the mission in its true sense. The robots took advantage of stigmergy to gradually repel each other, approach the walls, and eventually stop against them. The

evolutionary process tuned the timing of the behaviour to match the typical amount of time that elapsed between the beginning of the experiment and the moment when the blue signal appeared. This allowed the robots to score points by moving towards the walls before the appearance of the signal and remaining still against the walls after the appearance of the signal. Although this behaviour was reasonably well synchronised with the typical case, its failure to properly react to the appearance of the signal prevented it from achieving good scores consistently. Consequently, the performance achieved by `EvoPheromone` is significantly worse than the one achieved by both `Habanero` and `Human-Designers`.

## Aggregate results

To aggregate the performance of each design method across the four missions, we used a Friedman rank sum test on the performance observed in the real-robot experiments. The test indicates that, in the experiments presented, `Habanero` outranked all other design methods, with a confidence of at least 95%—see Fig. 5e. `Human-Designers` performed significantly better than both `EvoPheromone` and `Random-Walk`.

Figure 6 shows the aggregated execution time of the behaviour modules in the finite-state machines produced by `Habanero` and `Human-Designers`—measured in simulation. Results indicate that the finite-state machines produced by `Habanero` and `Human-Designers` are different: the execution time of the behaviour modules is different in `Habanero` and `Human-Designers` across all missions. Although `Habanero` and `Human-Designers` used the same set of modules, they combined them in a different way. The aggregated execution-time plot highlights four major differences between `Habanero` and `Human-`

`Designers`. First, `Habanero` used the Exploration module considerably less than `Human-Designers`. Second, `Habanero` relied more on modules that react to pheromone information compared to `Human-Designers`. Third, `Human-Designers` employed for a longer time the modules that respond to the walls' colour compared to `Habanero`. Finally, `Habanero` made greater use of the waggle module than `Human-Designers`.

While our experiments highlight performance differences between the two methods, we cannot definitively determine how the design choices made by `Habanero` and `Human-Designers` influence the overall performance. More precisely, our experimental setup cannot adequately explain the rationale behind the selection, tuning, and combination of the modules for either `Habanero` or `Human-Designers`, and its relationship with the performance obtained.

## Discussion

Automating the production of control software for pheromone-based robot swarms is a step further towards their real-world application. Automatic design can ease the realisation of robot swarms across different missions, while minimising human intervention[36,41,42,57]. The experiments presented in this paper show that this holds true also in the case of robot swarms that rely on pheromone-based stigmergy. Indeed, `Habanero` automatically designed stigmergy-based collective behaviours that were effective across all missions considered. For each mission, it found appropriate ways to use the pheromone effectively. Although the software modules on which `Habanero` operates were conceived in a mission-agnostic way, the interaction strategies that `Habanero` eventually generated for each mission were tailored to each of them and are different from one another. In these interaction strategies, the limited perception and computation capabilities of the individual robots are compensated at the swarm level by exploiting pheromone-based stigmergy. The e-puck used in the experiments, as a single robot, has limited spatial coordination, memory, and communication abilities. However, spatial organisation, external memory, and communication in the swarm emerged at the collective level thanks to pheromone-based stigmergy. Spatial organisation: In AGGREGATION, DECISION MAKING, and RENDEZVOUS POINT, the e-pucks self-organised and distributed in space guided by their pheromone trails and other environmental cues. Memory: In DECISION MAKING and RENDEZVOUS POINT, the swarm of e-pucks retained relevant information about the past state of the environment by laying pheromone trails. Communication: The semantics of pheromone trails is mission-specific. For example, the pheromone trails that the e-pucks laid in STOP had a meaning (stop where you are) that is radically different from the meaning in AGGREGATION (come here). It is interesting to note that spatial organisation, memory, and communication (including the semantics of pheromone trails) were not hand-coded in the modules on which `Habanero` operates: they were the product of the way in which `Habanero` automatically combined these modules on a per-mission basis.

The study leaves two main questions open. (i) Can automatic design leverage the intensity of pheromone trails and their decay time? In the experiments presented, a robot either did or did not sense the pheromone, in a binary fashion. A more thorough investigation is required to determine whether an automatic method can simultaneously tune the concentration of the pheromone deployed and the concentration to which a robot should react. (ii) Can automatic design methods realise robot swarms that alternatively, or simultaneously, operate with direct and indirect communication? We have shown in the past that direct communication can emerge from an automatic design process[53,58]. In this paper, we have shown that indirect communication can emerge as well. Further research is required to determine whether an automatic method can select direct or indirect communication as more suitable for a specific mission. In this sense, we deem particularly interesting the idea of automatically designing collective behaviours in which the robots operate with combinations of the two.

In this study, we adopted an existing technology to enable pheromone-based stigmergy with real robots—the photochromic artificial pheromone system[30]. Although viable, it is a technology that—like all the existing solutions—has some critical limitations: namely, it is only suitable for indoor applications in which the environment can be prepared beforehand with the photochromic material. As of today, no technology exists to provide robots with a universally applicable capability to mark their environment with the indication of their activities. However, by analysing the strengths of the available solutions, we can outline desirable properties for such technology. First, pheromones should be produced by robots, minimising the need for environment preparation and/or external infrastructure. Additionally, robots should have the ability to modulate the intensity of the pheromones they lay and respond to, enabling precise control over their behaviour. We also envision that pheromone-based stigmergy should facilitate the design of more complex behaviours, possibly by functioning over diverse types of pheromones that communicate different information. The devices that lay and sense pheromones should be easy to build and integrate in modern robot platforms at different scales—from small educational robots to larger platforms. Finally, the pheromone laid by the robots must be safe and nondestructive, and any marks left by the robots should disappear once the swarm completes its operation. Engineering solutions that meet these properties would facilitate their broad adoption, development, and validation, as well as the establishment of benchmarks for robotics stigmergy.

With `Habanero` we demonstrated that it is possible to generate pheromone-based collective behaviours through an automatic process that is repeatable and generally applicable. We contend that this result can motivate further research to overcome the limitations of the currently available hardware solutions to implement pheromone-based stigmergy.

## Methods

### Arena

All experiments were performed in a rectangular arena whose walls were realised with modular RGB blocks that display colours according to the mission requirements[53,59]—see Fig. 2b. The technical diagrams of the arenas used in the study are shown in Fig. 3. The floor of the arena was white and coated with a photochromic material that acts as a medium to encode the pheromone trails[30]. The coating was realised using an acrylic binder with a 20% (w/w) concentration of photochromic pigments. Technical information to reproduce the arena is provided as Supplementary Note 5. The photochromic material adopted turns magenta when exposed to UV light. Once the UV light is removed, the magenta colour gradually fades and the floor returns white in about 50 s—see Supplementary Video 5.

### The e-puck robot

The experiments were performed with e-puck robots—small-sized differential-drive robots that are widely adopted in swarm robotics research[52,60]. We used an extended version of the e-puck that is equipped with the Overo Gumstix computer-on-module to run Linux on the robot; the ground sensor module to detect the gray-level colour of the floor; a UV-light module and an omnidirectional camera to deposit and detect artificial pheromone trails, respectively. The UV-light module is a ring-shaped add-on module for e-puck that is equipped with nine down-facing UV LEDs positioned at the rear of the robot[30]. A picture of the hardware configuration of the e-puck robot adopted in the research is given in Fig. 2a. The capabilities of the e-puck for laying and detecting the artificial pheromone are illustrated in Supplementary Video 5.

**Reference model**: the extended version of e-puck adopted is described by reference model RM 4.1, which formally defines the input and output variables associated with sensors and actuators, respectively—see Fig. 2c. The control software of the robot reads/writes the input/output variables at every control step, which has a duration of 100 ms[61].

**Simulator**: all simulations were performed using ARGoS3 Version 48, along with the argos3-epuck-phormica library—see section Code Availability. ARGoS was specifically developed to simulate robot swarms[54]; the argos3-epuck-phormica library enables the cross-compilation of control software for the e-puck so that it can be ported to the robots without any manually applied modification.

**Table 1 | Habanero's low-level behaviours and transition conditions**

| Low-level behaviours | Parameters | Description |
|---|---|---|
| Exploration | $phe, \tau$ | Robot moves by random walk |
| Stop | $phe$ | Robot stops in place |
| Go-to-Colour | $phe, c, fov$ | Robot moves toward objects displaying a specific colour |
| Avoid-Colour | $phe, c, fov$ | Robot moves away from objects displaying a specific colour |
| Go-to-Pheromone | $phe, fov$ | Robot moves towards pheromone perceived in the surroundings |
| Avoid-Pheromone | $phe, fov$ | Robot moves away from pheromone perceived in the surroundings |
| Waggle | $phe$ | Robot rotates in place for a random period of time |
| **Transition conditions** | **Parameters** | **Description** |
| White-Floor | $\beta$ | White floor detected |
| Gray-Floor | $\beta$ | Gray floor detected |
| Black-Floor | $\beta$ | Black floor detected |
| Colour-Detected | $\beta, c, fov$ | Objects of a specific colour perceived |
| Pheromone-Detected | $\beta, fov$ | Pheromone detected in the surroundings |
| Fixed-Probability | $\beta$ | Transition with a fixed probability |

While performing all the low-level behaviours, the robot releases thin or thick pheromone trails if $phe$ is set to *thin* or *thick*, respectively. Otherwise, if $phe$ is set to *none*, the robot does not release a pheromone trail. The parameter $fov \in \{\frac{1}{12}\pi, 2\pi\}$ determines the field of view of the camera. The parameter $\tau \in \{1, \ldots, 100\}$ denotes the number of control steps for which a robot rotates in place while performing the exploration behaviour: a control cycle is 100 ms. The parameter $\beta \in [0, 1]$ determines the probability of transitioning in all transition conditions. The parameter $c \in \{R, G, B, C, Y\}$ denotes the colour to which the robots react when performing a particular behaviour or transition from colour-detected behaviour to another.

## Habanero

Habanero is an instance of AutoMoDe[40] specialised in the design of swarm of robots that can lay and detect pheromone trails. Habanero produces control software by assembling predefined software modules into probabilistic finite-state machines in which states are low-level behaviours performed by the robots and transitions are enabled by conditions on the contingencies experienced by the robot.

Habanero operates on seven low-level behaviours and six conditions. Both low-level behaviours and conditions have free parameters that affect their functioning. The space of solutions that Habanero can produce comprises all the possible probabilistic finite-state machines—with at most 4 states and at most 4 outgoing transitions per state—that can be obtained by assembling the available modules and by fine-tuning their free parameters. There are a total of 105 parameters to be tuned—with categorical parameters for the selection of software modules; and categorical, integer and real parameters that affect their functioning. The optimisation problem is mixed-variable in nature[62]. Habanero searches this space using Iterated F-race[55] with the goal of maximising a given mission-specific objective function. Iterated F-race samples, fine-tunes and selects candidate solutions performing simulations in ARGoS3. There is a limited number of simulations available to Habanero to produce an instance of control software—a simulations budget. Once the budget is exhausted, Habanero returns the best control software found up to that moment. A pictorial representation of Habanero is given in Fig. 1a.

The seven low-level behaviours are: exploration, stop, go-to-colour, avoid-colour, go-to-pheromone, avoid-pheromone, and waggle. The six conditions are: white-floor, gray-floor, black-floor, colour-detected, pheromone-detected, fixed-probability—see Fig. 1b,c and Table 1. All the low-level behaviours and the conditions interact with the e-puck hardware (sensors and actuators) via the input/output variables defined in reference model RM 4.1—see Fig. 2b.

We chose Iterated F-race to conduct Habanero's optimisation process as, for historical reasons, it is the de facto standard optimisation algorithm in the AutoMoDe family. Notably, Iterated F-race outperformed human experts in the modular design of control software for robot swarms[56]. Moreover, Iterated F-race was successful when applied to the problem of producing collective behaviours with a diverse set of AutoMoDe methods[40]. Iterated F-race has properties that make it suitable to tackle problems in the automatic modular design of control software. Particularly, it was conceived for the statistical selection of candidate solutions when (i) the problem instances are stochastic and (ii) the solutions comprise

discrete and continuous parameter spaces[55,63,64]. Recent studies have shown that other optimisation algorithms are suitable for the AutoMoDe family (e.g., simulated annealing[65] and sequential model-based algorithm configuration[66,67]). However, there is no evidence that indicates that they offer a definite advantage over Iterated F-race—see Kuckling[68] for a recent in-depth discussion.

## Comparisons

EvoPheromone is an adaptation of EvoStick, which is a standard neuroevolutionary method to design robot swarms[43]. EvoPheromone produces control software for an extended version of the e-puck robot formally described by reference model RM 4.1—same as Habanero. The architecture of the control software is a fully connected feed-forward artificial neural network. The neural network has 61 input nodes, 7 output nodes, and no hidden layer. The input and output nodes are directly connected by synaptic connections with weights. There are a total of 427 parameters to be tuned—all real values, which encode the synaptic weights. The optimisation problem is continuous in nature[62]. EvoPheromone tunes the synaptic weights of the neural network via elitism and mutation[43]. The evolutionary process is based on simulations executed in ARGoS3 with the argos3-epuck-phormica library—same setting as Habanero. The design process ends when a predefined simulation budget is exhausted. We developed EvoPheromone on the basis of EvoStick, as the latter is a readily available method for the e-puck that has served as a yardstick to apprise the performance of AutoMoDe methods in the past[43,56]. EvoStick is the only neuroevolutionary method that has been tested in the automatic design of robot swarms for several missions, without undergoing any mission-specific modification[48]. Moreover, EvoStick served as a starting point to develop other neuroevolutionary methods for robots endowed with enhanced capabilities—see, for example, adaptations of EvoStick to study direct communication[53,58] and spatial organisation[69]. EvoStick, and therefore EvoPheromone, are simple and straightforward implementations of the neuroevolutionary approach. We do not consider more advanced neuroevolutionary methods (e.g., CMA-ES[70], xNES[71], and NEAT[72]) as previous research has shown that they do not provide any performance advantage over EvoStick when applied off the shelf[48].

Human-Designers is a manual design method in which 10 human designers were requested to produce control software using the software modules of Habanero. In a sense, a human designer acts as an optimisation agent that assembles a finite-state machine and fine-tunes its parameters. Human-Designers produces control software for an extended

version of the e-puck robot formally described by reference model RM 4.1—same as `Habanero`. The human designers who participated in this study had various levels of expertise in swarm robotics—ranging from bachelor students to post-doctoral researchers in swarm robotics. Seven of them had previous experience with real robots, seven had previous experience with ARGoS3, and six had experience with the e-puck—either in simulation or reality. We provided the designers with a visualisation tool to produce and manipulate finite-state machines, to visualise simulations, and to compute the value of the objective function[73]. All simulations were executed in ARGoS3 with the argos3-epuck-phormica library—same setting as `Habanero`. The designers were allotted 4 hours per mission—see Supplementary Note 4. The guidelines and experimental description given to the designers are provided as Supplementary Note 3.

`Random-Walk`, although not an automatic design method, is included in the study as a lower bound on the performance of robot swarms. In `Random-Walk`, the robots move straight in the arena, when they encounter an obstacle, they rotate for a random number of control steps and then resume their straight motion. `Random-Walk` was conceived for an extended version of the e-puck robot formally described by reference model RM 4.1—same as `Habanero`.

## Missions

The empirical study is based on four missions. Each mission must be performed within $T = 180$ s by a swarm of $N = 8$ robots. The size of the swarm was determined in accordance with the number of robots available for the experiments.

**AGGREGATION**: initially, the robots are randomly placed in the arena—see Fig. 3a. The robots must approach one another to form a cluster and remain close until the end of the mission. Formally, the mission is specified by the following objective function, which must be minimised:

$$F_a = \sum_{t=1}^{T/100\,\text{ms}} d_{\text{avg}}(t). \tag{1}$$

At each control step $t$, the average distance $d_{\text{avg}}$ between the robots is added to $F_a$.

**DECISION MAKING**: initially, the robots are randomly placed in the arena—see Fig. 3b. The robots must select between a green and a blue region: at every control step $t$, the score is increase by $+1$ for every robot that is in the green region, and by $+2$ for every robot that is in the blue one. Both green and blue light signals disappear after a random amount of time, which is uniformly sampled between 70 and 90 s. Formally, the mission is specified by the following objective function, which must be maximised:

$$F_d = \sum_{t=1}^{T/100\,\text{ms}} \sum_{i=1}^{N} I_i(t); \qquad I_i(t) = \begin{cases} 1 & \text{if robot } i \text{ is in green region,} \\ 2 & \text{if robot } i \text{ is in blue region,} \\ 0 & \text{otherwise.} \end{cases} \tag{2}$$

**RENDEZVOUS POINT**: initially, the robots are placed in the left side of the arena. The robots must reach the green region and stay there until the end of the mission. A blue region is added as a decoy to possibly confuse the robots—see Fig. 3c. Both green and blue light signals disappear after a random amount of time, which is uniformly sampled between 70 and 90 s. Formally, the mission is specified by the following objective function, which must be maximised:

$$F_r = K_{\text{in}} - K_{\text{out}}; \tag{3}$$

where $K_{\text{in}}$ is the number of robots inside the green region at the end of the mission, and $K_{\text{out}}$ is the number of robots outside.

**STOP**: initially, the robots are randomly placed in the arena. A blue light signal appears after a random amount of time $\bar{t}$, which is uniformly sampled between 70 and 90 s—see Fig. 3d. All the robots must stop as soon as the signal appears, but not before. Formally, the mission is specified by the

following objective function, which must be minimised:

$$F_s = \sum_{t=1}^{\bar{t}} \sum_{i=1}^{N} \bar{I}_i(t) + \sum_{t=\bar{t}+1}^{T} \sum_{i=1}^{N} I_i(t); \qquad I_i(t) = \begin{cases} 1 & \text{if robot } i \text{ is moving,} \\ 0 & \text{otherwise;} \end{cases} \qquad \bar{I}_i(t) = 1 - I_i(t). \tag{4}$$

In the absence of well-established benchmark missions, we chose a set of missions that allowed us to estimate the expected performance of `Habanero` in typical swarm robotics tasks. AGGREGATION, DECISION MAKING, RENDEZVOUS POINT and STOP are missions that belong into the same class—they allow the pheromone-based coordination of robots. Yet, they are sufficiently different to benefit from a tailored design—they vary in the nature of their goals and in the presence of reference points of interest. By selecting a varied set of missions, we also aimed at testing `Habanero`'s ability to handle diverse challenges without undergoing any mission-specific adjustment.

It is worth noting that these missions—likewise `Habanero`—are not suitable for drawing conclusions on whether automatic methods can handle more complex missions or design relatively more complex stigmergy-based interactions. For instance, missions that require precise behavioural control via careful modulation of the pheromone deposition and response, or missions that involve more complex communication strategies through various types of pheromones.

## Protocol

All experiments were executed without any human intervention or any mission-specific modification in the design process. In the case of `Habanero` and `EvoPheromone`, for each mission, we independently executed the design process 10 times to obtain 10 instances of control software. Both methods operated with a budget of 100,000 simulation runs for each execution of the design process. We executed all automatic design processes on a high-performance computational cluster with about 1500 computing cores. In case of `Human-Designers`, 10 human designers were involved and each of them produced one instance of control software for each mission. After obtaining all the instances of control software, we assessed their performance once in simulation and once in reality. We varied the initial position of the robots when assessing instances of control software of a single method, and we used the same set of initial positions across the four methods. To perform the experiments in reality, the instances of control software, regardless of the design method that produced them, were automatically cross-compiled and deployed on the e-puck robots without undergoing any manually-applied modification.

**Tracking system.** We used a tracking system to automatically compute the performance of a robot swarm during each run of a real-robot experiment[74]. The tracking system uses an overhead camera to record the positions of the robots by recognising squared markers mounted on the robots. We also used the overhead camera to record videos of the experiments—see Supplementary Video 6. The overhead camera was used only to measure the performance of the swarm and was not used to provide any information to the robots.

## Statistics

We present the performance of the different methods with notched box-and-whiskers plots on a per-mission basis. In these plots, boxes represent the interquartile range, covering the central 50% of the values observed. Whiskers extend from the lower quartile to the lowest recorded performance, and from the upper quartile to the highest one. The horizontal line in the middle of each box plot represents the median performance, and the notches on the box represent a 95% confidence interval on the median. If the notches of two boxes do not overlap, then the difference between their respective medians is significant, with a confidence of at least 95%[75]. For each method, we present the performance obtained in simulation and in

real-robot experiments using thin and thick boxes, respectively. We executed a mission-specific comparison of the performance of methods with Wilcoxon paired rank sum tests at 95% confidence[76].

We also performed a Friedman rank sum test[76] that aggregates the performance of each method across all four missions. More precisely, we applied a Friedman two-way analysis of variance to the performances recorded in the experiments with physical robots, across all missions, and for all methods. The Friedman test is nonparametric and implements a block design. In our protocol, the treatment factor is the method under analysis and the blocking factor is the mission. By operating on the ranks, the Friedman test is invariant to the magnitude of the objective functions of the missions considered. Also, due to its nonparametric nature, it can be applied with no assumption on the distribution of the performance. These properties are instrumental for aggregating the performance observed across the four missions. We present the results of the test with the average rank of each method (computed across all missions), and its 95% confidence interval. A method is significantly better than other if it has a lower average rank and the confidence interval of the two methods do not overlap.

## Data availability

The data that support the findings of this study are available in figshare with the identifier https://doi.org/10.6084/m9.figshare.24707154.

## Code availability

The software used to produce the results of our study is available in the following public repositories under the MIT License: ARGoS3 (https://doi.org/10.5281/zenodo.4889111) for the ARGoS3 simulator; irace (https://doi.org/10.5281/zenodo.4888996) for the implementation of Iterated F-race; ARGoS3-AutoMoDe (https://doi.org/10.5281/zenodo.7090227) for the implementation of Habanero and Random-Walk; demiurge-epuck-dao (https://doi.org/10.5281/zenodo.7150581) for the reference model of the robots used by all design methods; experiments-loop-functions (https://doi.org/10.5281/zenodo.7150584) for the objective functions used to compute the score in a mission in simulation; argos3-epuck-private (https://doi.org/10.5281/zenodo.7241397) ARGoS3 plugin for the e-puck robot endowed with the UV module; argos3-phormica (https://doi.org/10.5281/zenodo.7241409) ARGoS3 plugin to enable the simulation of Phormica—a pheromone release and detection system; ARGoS3-NEAT (https://doi.org/10.5281/zenodo.7150530) for the implementation of EvoPheromone; experiments-loop-functions-ros (https://doi.org/10.5281/zenodo.7241441) plugin to compute the performance score in real robot experiments; and AutoMoDe-visualisation-tool (https://doi.org/10.5281/zenodo.7241468) a web editor tool that allows Human-Designers to manually edit Auto-MoDe finite-state machines and visualise its performance in simulation. Installation and execution instructions are provided as Source Data.

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

## Acknowledgements

We thank the students and the researchers that participated in the study (`Human-Designers`). We also thank Dr. Mary Katherine Heinrich and Dr. Carlo Pinciroli for reading and commenting on a preliminary version of the paper. The project has received funding from the European Research Council (ERC) under the European Union's Horizon 2020 research and innovation programme (DEMIURGE Project, grant agreement No 681872), from Belgium's Wallonia-Brussels Federation through the ARC Advanced project GbO–Guaranteed by Optimization, and from the Belgian Fonds de la Recherche Scientifique–FNRS through the crédit d'équipement SwarmSim. DGR acknowledges support from the Colombian Ministry of Science, Technology and Innovation–Minciencias. MB acknowledges support from the Belgian Fonds de la Recherche Scientifique–FNRS.

## Author contributions

The three authors developed the original ideas, defined the methodology, and contributed to the provision of the resources. M.S. and D.G.R developed the software, conducted the experiments, gathered and visualised empirical data, and drafted the initial version of the manuscript. The three authors validated the research outputs, and formally analysed the results. Together, they wrote revised and edited the final version of the manuscript. MB acquired the funding, supervised the research, and managed the project. M.S. and D.G.R. contributed equally to this work.

## Competing interests

The authors declare no competing interests.
