## [Peer Review File · Communications Engineering]

Reviewers' comments:

Reviewer #1 (Remarks to the Author):

This paper proposes an automatic design method for stigmergy-based behaviors of swarm robotic system, called Habanero. Leveraging the existing software and hardware setup for a stigmergy-based swarm robotic system, the authors demonstrated the proposed method compared to the three baseline methods: EvoPheromone, Human-Designers and Random Walk. Habanero demonstrated its effectiveness in designing stigmergy-based behaviors over four different tasks.

Reviewer's Comment:

- The author provided a very convincing and clear introduction stating why automatic design methods need to be developed in stigmergy-based swarm robotic behaviors, mentioning highly relevant previous studies.

- In the caption of Figure 2, it seems the description (b) and (c) are swapped. Please check it carefully and provide an accurate description for each figure.

- The proposed Habanero Algorithm uses the Iterated F-race algorithm to fine-tune the free parameters to provide the best probabilistic combination of seven low-level behaviors and six conditions. Why is the iterated F-race algorithm specifically chosen to optimize the parameters? What is the advantage of the iterated F-race algorithm over other optimization algorithms?

- In the supplementary videos, the EvoPheromone method showed undesirable behaviors which are critical for the robot swarm to perform the given tasks. For example, when the robot swarm was performing aggregation and decision-making using the EvoPheromone, most of the robots were stuck on the wall until the end of the experiment. These undesirable behaviors significantly dropped the performance metric of the EvoPheromone, which was used to evaluate the performance of the Habanero method. Although it is the limitation of the EvoPheromone, it could have been easily mitigated with an improvement in the method. Due to that, the performance of the automatic design aspect of the Habanero method cannot be directly compared with the method used for neuroevolution. Please provide justification of the selection of the original EvoPheromone as the baseline without any modification to prevent any undesirable behaviors not directly related to the given tasks, but affecting the task performance significantly.

- The advantage of the Habanero method is shown clearly when simulation and real robot experiments are compared. In Figure 5, the quantitative measure to compare the performance of each method is given. To help the readers to understand how the Habanero method helped to narrow down the sim-to-real gap, please provide a quantitative metric of the sim-to-real gap for each method for the task and use the metric as direct supporting result for smaller sim-to-real gap using the Habanero method.

Reviewer #2 (Remarks to the Author):

The paper presents an approach for automatically designing collective behavior for pheromone-based robot swarms. The approach is novel and has been well-performed with four different missions for both simulation and real-world implementation. The manuscript is clear and very interesting to the swarm robotics community as it deals with the challenge of designing fully autonomous and distributed collective controllers, especially for pheromone-based robot swarms. Overall, the approach seems promising, and the experimental results support its efficacy. However, I would like to raise some concerns that could be addressed:

The author claims on page 2 that UV light-based artificial pheromones still require prepared environments, limiting the flexibility of robot swarms. It would be helpful to see some discussion on potential solutions to designing perfect artificial pheromones.

On page 2, another method uses deep reinforcement learning (DRL) for automatic control design with a virtual pheromone. It would be more convincing to compare the proposed Habanero method's design efficiency and performance with DRL.

On page 2, is there a possibility that the reality gap cannot be crossed? For example, will the design space and variance become larger when given more low-level behaviors and state transitions?

On page 3, the paper uses Iterated F-race as the optimization algorithm. It would be helpful to explain the advantages of using this algorithm and address any potential issues, such as local minimum problems.

On page 4, EvoPheromone works in simulation but failed in experiments. It would be helpful to discuss the potential reasons why the behavior could not be reproduced in the real world.

On page 5, since decision making requires 'memory', it would be helpful to use memory neural network architecture in EvoPheromone, such as replacing the fully connected network with long short-term memory networks, and compare the performance.

On page 8, it is unclear how many simulation tests were conducted to show the performance distribution in figure 5. Were each simulation with random robot positions? Were all four methods tested on the same simulation conditions?

On page 8, the Friedman rank sum test was used to show performance across the four missions. It would be helpful to explicitly explain the test's meaning, particularly whether the objective function combines the four mission objectives.

On page 9, Habanero and Human-Designers combine low-level behaviors in a different way, as shown in Figure 6. It would be helpful to provide an analysis of the difference between the combination ways and explain why Habanero is better than Human-Designers.

On page 11, it would be helpful to know the number and type of free parameters used in the optimization process and how they might influence the optimization results.

Other small notes:

- * Page 2: "pheromone trail on ground . that has".
- * Page 12: "although not a automatic design".

Reviewer #3 (Remarks to the Author):

This manuscript deals with stigmergy-based robot swarms, and more specifically with strategies to automatically design the 'controller', i.e. the local rules leading to an emergent collective behavior solving a given macro-problem. This work combines simulations and actual experiments with 8 robots (modified e-pucks). There are very few swarm robotic experiments involving stigmergic systems; this in itself makes this study noteworthy and valuable to our community. The stigmergy is based on artificial pheromone trails that are laid using UV light.

Overall the manuscript is well written and the research extensive. The research question is clearly stated and motivated. It is indeed an important practical problem; maybe the most critical issue when designing swarm robotic systems using the environment for its coordination. Specifically, the authors present a new strategy call 'Habanero', which is an automatic offline design method that belongs to the AutoMoDe family, previously developed by some of the authors. In that respect, this work is an extension of this previous line of work.

The results are gathered for 4 missions (aggregation, decision-making, rendezvous point, stop). They are quite compelling.

Having said that, I have a number of questions/comments (I may have missed some information while reading the manuscript) that have to be addressed:

- * Why so few robots are used (N=8)? I suspect this is limited by the number of units available for the experiments. However, the size of the swarm (or equivalently its density) plays a fundamental role and some recent articles (for non stigmergic systems) extensively studied the emergence of swarm behaviors as a function of the size/density.
- * In relation with the previous point: if the constraint on the size of the system is purely experimental, why not performing simulations with a larger system? This point must be discussed.
- * The results shown in Figure 5 can be extremely confusing at first, given that in some cases the objective function is minimized, while in other cases it's maximized. I strongly recommend to be consistent and have only minimization or maximization.
- * I may have missed it but how many times are repeated each mission to obtain the box plots?
- * This manuscript partially addresses a general and important issue in swarm robotics: the lack of benchmark. Maybe the authors could discuss what they think could be done to deal with this critical issue.
- * I've noticed the presence of an overhead camera to track the e-puck positions. Could the authors confirm that this camera is only used to measure the positions in order to quantify the performance (and not for the actual positioning and used by the control algorithm)?
- * I was intrigued by the performance of Overo Gumstick single-board-computer. Was it sufficient to carry out all the onboard simulations?

* The 4 missions considered by the authors are fairly standard. I was wondering if the authors anticipate any issue using Habanero for a more complex mission. What are the limitations?

Answers to the Reviewers

August 28, 2023

We would like to express our gratitude to the reviewers for their insightful comments and suggestions. We believe that these revisions have greatly improved the quality of our work. Please find below our point-by-point responses. The text in blue are paragraphs that we have added to the manuscript in this revision to address the comments of the reviewers.

Reviewer 1

1. **Comment:** This paper proposes an automatic design method for stigmergy-based behaviors of swarm robotic system, called Habanero. Leveraging the existing software and hardware setup for a stigmergy-based swarm robotic system, the authors demonstrated the proposed method compared to the three baseline methods: EvoPheromone, Human-Designers and Random Walk. Habanero demonstrated its effectiveness in designing stigmergy-based behaviors over four different tasks. The author provided a very convincing and clear introduction stating why automatic design methods need to be developed in stigmergy-based swarm robotic behaviors, mentioning highly relevant previous studies.

Answer: We thank the reviewer for the positive comment.

2. **Comment:** In the caption of Figure 2, it seems the description (b) and (c) are swapped. Please check it carefully and provide an accurate description for each figure.

Answer: We corrected the caption.

(b) The reference model RM 4.1, which formally describes the interface between the robot and the control software. (c) The experimental arena. The floor is coated with photochromic material. It changes in color from white to magenta when exposed to UV light, and gradually returns to its normal white color when UV light is removed.

3. **Comment:** The proposed Habanero Algorithm uses the Iterated F-race algorithm to fine-tune the free parameters to provide the best probabilistic combination of seven low-level behaviors and six conditions. Why is

the iterated F-race algorithm specifically chosen to optimize the parameters? What is the advantage of the iterated F-race algorithm over other optimization algorithms?

Answer: We clarified the rationale behind choosing Iterated F-race.

We chose Iterated F-race to conduct *Habanero*'s optimization process as, for historical reasons, it is the *de facto* standard optimization algorithm in the AutoMoDe family. Notably, Iterated F-race was the optimization algorithm to firstly outperform human experts in the modular design of control software for robot swarms [57]. Moreover, Iterated F-race was successful when applied to the problem of producing collective behaviors with a diverse set of AutoMoDe methods [40]. Iterated F-race has properties that make it suitable to tackle problems in the automatic modular design of control software. Particularly, it was conceived for the statistical selection of candidate solutions when (i) the problem instances are stochastic and (ii) the solutions comprise discrete and continuous parameter spaces [56, 64, 65]. Recent studies have shown that other optimization algorithms are suitable for the AutoMoDe family (e.g., simulated annealing [66] and sequential model-based algorithm configuration [67, 68]). However, there is no evidence that indicates that they offer a definite advantage over Iterated F-race—see Kuckling [69] for a recent in-depth discussion.

4. **Comment:** In the supplementary videos, the *EvoPheromone* method showed undesirable behaviors which are critical for the robot swarm to perform the given tasks. For example, when the robot swarm was performing aggregation and decision-making using the *EvoPheromone*, most of the robots were stuck on the wall until the end of the experiment. These undesirable behaviors significantly dropped the performance metric of the *EvoPheromone*, which was used to evaluate the performance of the *Habanero* method. Although it is the limitation of the *EvoPheromone*, it could have been easily mitigated with an improvement in the method. Due to that, the performance of the automatic design aspect of the *Habanero* method cannot be directly compared with the method used for neuroevolution. Please provide justification of the selection of the original *EvoPheromone* as the baseline without any modification to prevent any undesirable behaviors not directly related to the given tasks, but affecting the task performance significantly.

Answer: We made explicit our motivation for developing *EvoPheromone* on the basis of *EvoStick*. We also point to literature that has shown that more advanced neuroevolutionary methods do not provide an advantage over the simpler *EvoStick* when used off the shelf.

We agree with the reviewer that there might exist possible strategies to steer the evolution of the neural networks toward behaviors that appear

desirable to perform some of the missions. However, researching and implementing those mission-specific modifications falls out of our vision for a fully-automatic design process.

In the absence of a well-defined state of the art, **EvoPheromone** is a suitable comparison yardstick. We believe that developing a generally-applicable neuroevolutionary method that can produce stigmergy-based behaviors and that performs well both in simulation and reality would be in itself great contribution to our field, and deserves an independent research work. We make available all our software and materials, so that interested researchers can use **EvoPheromone** as a starting point.

We developed **EvoPheromone** on the basis of **EvoStick**, as the latter is a readily available method for the e-puck that has served as a yardstick to apprise the performance of AutoMoDe methods in the past [43, 57]. **EvoStick** is the only neuroevolutionary method that has been tested in the automatic design of robot swarms for several missions, without undergoing any mission-specific modification [48]. Moreover, **EvoStick** served as a starting point to develop other neuroevolutionary methods for robots endowed with enhanced capabilities—see, for example, adaptations of **EvoStick** to study direct communication [59, 54] and spatial organization [70]. **EvoStick**, and therefore **EvoPheromone**, are simple and straightforward implementations of the neuroevolutionary approach. We do not consider more advanced neuroevolutionary methods (e.g., **CMA-ES** [71], **xNES** [72], and **NEAT** [73]) as previous research has shown that they do not provide any performance advantage over **EvoStick** when applied off the shelf [48].

5. **Comment:** The advantage of the Habanero method is shown clearly when simulation and real robot experiments are compared. In Figure 5, the quantitative measure to compare the performance of each method is given. To help the readers to understand how the Habanero method helped to narrow down the sim-to-real gap, please provide a quantitative metric of the sim-to-real gap for each method for the task and use the metric as direct supporting result for smaller sim-to-real gap using the Habanero method.

Answer: We provide now comparative results on the observed performance drop between simulation and reality. The results are available in Supplementary Document titled "robustness to the reality gap".

Reviewer 2

1. **Comment:** The paper presents an approach for automatically designing collective behavior for pheromone-based robot swarms. The approach is

novel and has been well-performed with four different missions for both simulation and real-world implementation. The manuscript is clear and very interesting to the swarm robotics community as it deals with the challenge of designing fully autonomous and distributed collective controllers, especially for pheromone-based robot swarms. Overall, the approach seems promising, and the experimental results support its efficacy. However, I would like to raise some concerns that could be addressed:

Answer: We thank the reviewer for the positive feedback.

2. **Comment:** The author claims on page 2 that UV light-based artificial pheromones still require prepared environments, limiting the flexibility of robot swarms. It would be helpful to see some discussion on potential solutions to designing perfect artificial pheromones.

Answer: With our study, we cannot support claims on technologies that could (in the future) tackle the limitations of current approaches to produce pheromone-based stigmergy. However, by taking into account the current technologies and their limitations, we indeed can identify desirable properties for better potential solutions. We extended the Discussion by describing these desirable properties.

As of today, no technology exists to provide robots with a universally applicable capability to mark their environment with indication of their activities. However, by analyzing the strengths of the available solutions, we can outline desirable properties for such technology. First, pheromones should be produced by robots, minimizing the need for environment preparation and/or external infrastructure. Additionally, robots should have the ability to modulate the intensity of the pheromones they lay and respond to, enabling precise control over their behavior. We also envision that pheromone-based stigmergy should facilitate the design of more complex behaviors, possibly by functioning over diverse types of pheromones that communicate different information. The devices that lay and sense pheromones should be easy to build and integrate in modern robot platforms at different scales—from small educational robots to larger platforms. Finally, the pheromone laid by the robots must be safe and non-destructive, and any marks left by the robots should disappear once the swarm completes its operation. Engineering solutions that meet these properties would facilitate their broad adoption, development, and validation, as well as the the establishment of benchmarks for robotics stigmergy.

3. **Comment:** On page 2, another method uses deep reinforcement learning (DRL) for automatic control design with a virtual pheromone. It would be more convincing to compare the proposed Habanero method’s design efficiency and performance with DRL.

Answer: The referred deep reinforcement learning method is indeed capable of designing pheromone-based stigmergy behaviors. However, during run-time, the method requires a centralized computation infrastructure to store global pheromone information and make it available to the robots. In *Habanero*, the robots lay and perceive pheromones with no need of external computation infrastructure. The two methods are not intended to be used in the same scenarios. Therefore, they operate under different working hypothesis and cannot be compared directly. We extended the description of the deep reinforcement learning method to clarify this point.

The only exception to manual design is one study in which deep reinforcement learning was used to develop a collision avoidance behavior based on a virtual pheromone [39]. Although restricted to simulation-only experiments, this study showed that control software produced through deep reinforcement learning can outperform the one generated via manual design. The proposed approach was conceived for scenarios where a centralized infrastructure stores global pheromone information and makes it accessible to the robots. On the one hand, this approach provides a solution to the problem of designing pheromone-based behaviors in virtual environments. On the other hand, the approach is not directly applicable in scenarios where the robots are expected to autonomously lay and sense the artificial pheromones in their physical environment.

4. **Comment:** On page 2, is there a possibility that the reality gap cannot be crossed? For example, will the design space and variance become larger when given more low-level behaviors and state transitions?

Answer: We rephrased this part to express that crossing the reality gap is something that happens in a degree ("more or less satisfactorily"), and not in a binary fashion ("successfully or unsuccessfully").

In the past, we observed that the size of the search space is mostly determined by the control architecture (number of possible combinations), and in a lesser extent by the number of pre-defined software modules (number of elements to be combined). We indicate now that the restricted architecture is the main factor.

This improvement can be attributed to AutoMoDe's constraint that control software must be generated by assembling the given modules within a specific architecture (e.g., a probabilistic finite-state machine). By applying this constraint, AutoMoDe limits the size of the design space to the set of possible combinations of modules, and therefore reduces the variance of the design process [43]. This reduces the risk of over-fitting the control

software produced to the idiosyncrasies of the simulation environment, which is the main reason why control software might fail to cross the reality gap satisfactorily [47].

5. **Comment:** On page 3, the paper uses Iterated F-race as the optimization algorithm. It would be helpful to explain the advantages of using this algorithm and address any potential issues, such as local minimum problems.

Answer: We clarified the rationale behind choosing Iterated F-race.

In *Habanero*, Iterated F-race operates as a black-box optimization algorithm. This means that, at design time, the algorithm does not make assumptions on the optimization landscape and/or on the existence of single optima. Iterated F-race is an optimization algorithm that we use to find suitable instances of control software, without optimality guarantees.

We chose Iterated F-race to conduct *Habanero*'s optimization process as, for historical reasons, it is the *de facto* standard optimization algorithm in the AutoMoDe family. Notably, Iterated F-race was the optimization algorithm to firstly outperform human experts in the modular design of control software for robot swarms [57]. After that, Iterated F-race was successful when applied to the problem of producing collective behaviors with a diverse set of AutoMoDe methods [40]. Iterated F-race has properties that make it suitable to tackle problems in the automatic modular design of control software. Particularly, it was conceived for the statistical selection of candidate solutions when (i) the problem instances are stochastic and (ii) the solutions comprise discrete and continuous parameter spaces [56, 64, 65]. Recent studies have shown that other optimization algorithms are suitable for the AutoMoDe family (e.g., simulated annealing [66] and sequential model-based algorithm configuration [67, 68]). However, there is no evidence that indicates that they offer a definite advantage over Iterated F-race—see Kuckling [69] for a recent in-depth discussion.

6. **Comment:** On page 4, EvoPheromone works in simulation but failed in experiments. It would be helpful to discuss the potential reasons why the behavior could not be reproduced in the real world.

Answer: In the manuscript, we describe to behaviors observed in simulation that we believe did not materialize effectively in the physical swarm. We describe these behaviors in the Results section, on a per-mission basis. However, with our experiments, we cannot convincingly isolate the reasons for which these behaviors do not transfer well. We believe the main reason for the observed behavior are the known effects of the reality gap in neuroevolutionary methods. We have extended our study on the

effects of the reality gap in a new Supplementary Document titled "robustness to the reality gap". In the document, we also provide references to previous work that discusses in-depth such effects, both in modular and neuroevolutionary methods.

7. **Comment:** On page 5, since decision making requires 'memory', it would be helpful to use memory neural network architecture in `EvoPheromone`, such as replacing the fully connected network with long short-term memory networks, and compare the performance.

Answer: We extended the manuscript with a more detailed motivation for conceiving `EvoPheromone` on the basis of the simple neuroevolutionary method `EvoStick`. Even though there are more advanced architectures, past research has shown that they do not necessarily provide an advantage when used off the shelf.

In this experimental setup, the swarm can indeed benefit from a memory behavior to perform Decision Making. However, we can imagine also alternative behaviors that do not require memory. For example, moving towards the colored walls and stop and stand still when sensing objects in the surroundings.

We agree with the reviewer that many possible paths could be taken to harness the full representative power of neural network architectures, and produce more advanced pheromone-based stigmergy. However, we believe this is a research work that would be a contribution on its own and we could not possibly address it in the current manuscript.

We developed `EvoPheromone` on the basis of `EvoStick`, as the latter is a readily available method for the e-puck that has served as a yardstick to apprise the performance of `AutoMoDe` methods in the past [43, 57]. `EvoStick` is the only neuroevolutionary method that has been tested in the automatic design of robot swarms for several missions, without undergoing any mission-specific modification [48]. Moreover, `EvoStick` served as a starting point to develop other neuroevolutionary methods for robots endowed with enhanced capabilities—see, for example, adaptations of `EvoStick` to study direct communication [59, 54] and spatial organization [70]. `EvoStick`, and therefore `EvoPheromone`, are simple and straightforward implementations of the neuroevolutionary approach. We do not consider more advanced neuroevolutionary methods (e.g., `CMA-ES` [71], `xNES` [72], and `NEAT` [73]) as previous research has shown that they do not provide any performance advantage over `EvoStick` when applied off the shelf [48].

8. **Comment:** On page 8, it is unclear how many simulation tests were conducted to show the performance distribution in figure 5. Were each

simulation with random robot positions? Were all four methods tested on the same simulation conditions?

Answer: We now provide more information on the experimental protocol. Both in Figure 5 and in the Methods.

We report results of the evaluation of 160 instances of control software, 10 per method and per mission. All instances of control software were evaluated once in simulation and once with physical robots—more details on the protocol are provided in Methods.

We varied the initial position of the robots when assessing instances of control software of a single method, and we used the same set of initial positions across the four methods.

9. **Comment:** On page 8, the Friedman rank sum test was used to show performance across the four missions. It would be helpful to explicitly explain the test’s meaning, particularly whether the objective function combines the four mission objectives.

Answer: We provide more details on the statistical analysis conducted via the Friedman test.

We also performed a Friedman rank sum test [77] that aggregates the performance of each method across all four missions. More precisely, we applied a Friedman two-way analysis of variance to the performances recorded in the experiments with physical robots, across all missions, and for all methods. The Friedman test is nonparametric and implements a block design. In our protocol, the treatment factor is the method under analysis and the blocking factor is the mission. By operating on the ranks, the Friedman test is invariant to the magnitude of the objective functions of the missions considered. Also, due to its nonparametric nature, it can be applied with no assumption on the distribution of the performance. These properties are instrumental for aggregating the performance observed across the four missions. We present the results of the test with the average rank of each method (computed across all missions), and its 95% confidence interval. A method is significantly better than other if it has a lower average rank and the confidence interval of the two methods do not overlap.

10. **Comment:** On page 9, Habanero and Human-Designers combine low-level behaviors in a different way, as shown in Figure 6. It would be helpful to provide an analysis of the difference between the combination ways and explain why Habanero is better than Human-Designers.

Answer: We now comment on information that we can extract from the aggregated execution-time plot. Unfortunately, our experimental setup was not designed to provide information about the rationale that Iterated F-race and the participants follow while combining the software modules. Therefore, we do not feel we can comment on why one combination might be better than other. We made explicit this limitation in the manuscript.

The aggregated execution-time plot highlights four major differences between **Habanero** and **Human-Designers**. First, **Habanero** used the **Exploration** module considerably less than **Human-Designers**. Second, **Habanero** relied more on modules that react to pheromone information compared to **Human-Designers**. Third, **Human-Designers** employed for a longer time the modules that respond to the walls' color compared to **Habanero**. Finally, **Habanero** made greater use of the **Waggle** module than **Human-Designers**.

While our experiments highlight performance differences between the two methods, we cannot definitively determine how the design choices made by **Habanero** and **Human-Designers** influence the overall performance. More precisely, our experimental setup cannot adequately explain the rationale behind the selection, tuning, and combination of the modules for either **Habanero** or **Human-Designers**, and its relationship with the performance obtained.

11. **Comment:** On page 11, it would be helpful to know the number and type of free parameters used in the optimization process and how they might influence the optimization results.

Answer: We now indicate the number and type of free parameters—both for **Habanero** and **EvoPheromone**. The number and type of parameters define the type of optimization problem addressed, which to some extent is an indicator of the possible challenges one might face while conducting optimization-based design processes. In this study, **Habanero** and **EvoPheromone** are methods that operate in a complete different way—both because of their design space and the optimization algorithm they use. Unfortunately, with our study, we cannot isolate the possible effects of the parameter space in the obtained results.

There are a total of 105 parameters to be tuned—with categorical parameters for the selection of software modules; and categorical, integer and real parameters that affect their functioning. The optimization problem is mixed-variable in nature [63].

There are a total of 427 parameters to be tuned—all real values, which encode the synaptic weights. The optimization problem is continuous in nature [63].

12. **Comment:** * Page 2: “pheromone trail on ground . that has”.

Answer: We corrected the sentence.

artificial pheromone trail on ground that has previously been coated with

13. **Comment:** * Page 12: “although not a automatic design”.

Answer: We corrected the sentence.

although not an automatic design method

Reviewer 3

1. **Comment:** “This manuscript deals with stigmergy-based robot swarms, and more specifically with strategies to automatically design the “controller”, i.e. the local rules leading to an emergent collective behavior solving a given macro-problem. This work combines simulations and actual experiments with 8 robots (modified e-pucks). There are very few swarm robotic experiments involving stigmergic systems; this in itself makes this study noteworthy and valuable to our community. The stigmergy is based on artificial pheromone trails that are laid using UV light. Overall the manuscript is well written and the research extensive. The research question is clearly stated and motivated. It is indeed an important practical problem; maybe the most critical issue when designing swarm robotic systems using the environment for its coordination. Specifically, the authors present a new strategy call “Habanero”, which is an automatic offline design method that belongs to the AutoMoDe family, previously developed by some of the authors. In that respect, this work is an extension of this previous line of work. The results are gathered for 4 missions (aggregation, decision-making, rendezvous point, stop). They are quite compelling. Having said that, I have a number of questions/comments (I may have missed some information while reading the manuscript) that have to be addressed:”

Answer: We thank the reviewer for the positive comments.

2. **Comment:** Why so few robots are used (N=8)? I suspect this is limited by the number of units available for the experiments. However, the size of the swarm (or equivalently its density) plays a fundamental role and some recent articles (for non stigmergic systems) extensively studied the emergence of swarm behaviors as a function of the size/density.

Answer: Indeed, the number of robots was an experimental limitation. We clarified this.

The size of the swarm was determined in accordance with the number of robots available for the experiments.

3. **Comment:** In relation with the previous point: if the constraint on the size of the system is purely experimental, why not performing simulations with a larger system? This point must be discussed.

Answer: We added an analysis of the scalability properties of the control software produced with *Habanero*. The analysis is available in Supplementary Document titled "habanero scalability analysis".

4. **Comment:** The results shown in Figure 5 can be extremely confusing at first, given that in some cases the objective function is minimized, while in other cases it's maximized. I strongly recommend to be consistent and have only minimization or maximization.

Answer: We refactored the plots as suggested by the reviewer.

5. **Comment:** I may have missed it but how many times are repeated each mission to obtain the box plots?

Answer: We clarified this in Figure 5 and in the Methods.

We report results of the evaluation of 160 instances of control software, 10 per method and per mission. All instances of control software were evaluated once in simulation and once with physical robots—more details on the protocol are provided in Methods.

We varied the initial position of the robots when assessing instances of control software of a single method, and we used the same set of initial positions across the four methods.

6. **Comment:** This manuscript partially addresses a general and important issue in swarm robotics: the lack of benchmark. Maybe the authors could discuss what they think could be done to deal with this critical issue.

Answer: Research work on robotics pheromone-based stigmergy is still sparse and remains disconnected across research groups. We line our vision on desirable properties for these technologies, as a way to start a discussion on possible goals that should be achieved by the research community. We believe benchmarks must arise from the collective incremental development of robotics technologies. We make this explicit in the paper.

As of today, no technology exists to provide robots with a universally applicable capability to mark their environment with indication of their activities. However, by analyzing the strengths of the available solutions, we can outline desirable properties for such technology. First, pheromones should be produced by robots, minimizing the need for environment preparation and/or external infrastructure. Additionally, robots should have the ability to modulate the intensity of the pheromones they lay and respond to, enabling precise control over their behavior. We also

envision that pheromone-based stigmergy should facilitate the design of more complex behaviors, possibly by functioning over diverse types of pheromones that communicate different information. The devices that lay and sense pheromones should be easy to build and integrate in modern robot platforms at different scales—from small educational robots to larger platforms. Finally, the pheromone laid by the robots must be safe and non-destructive, and any marks left by the robots should disappear once the swarm completes its operation. Engineering solutions that meet these properties would facilitate their broad adoption, development, and validation, as well as the the establishment of benchmarks for robotics stigmergy.

We also describe our approach for working in the absence of established benchmark problems.

In the absence of well-established benchmark missions, we chose a set of missions that allowed us to estimate the expected performance of *Habanero* in typical swarm robotics tasks. AGGREGATION, DECISION MAKING, RENDEZVOUS POINT and STOP are missions that belong into the same class—they allow the pheromone-based coordination of robots. Yet, they are sufficiently different to benefit from a tailored design—they vary in the nature of their goals and in the presence of reference points of interest. By selecting a varied set of missions, we also aimed at testing *Habanero*'s ability to handle diverse challenges without undergoing any mission-specific adjustment.

It is worth noting that these missions—likewise *Habanero*—are not suitable for drawing conclusions on whether automatic methods can handle more complex missions or design relatively more complex stigmergy-based interactions. For instance, missions that require precise behavioral control via careful modulation of the pheromone deposition and response, or missions that involve more complex communication strategies through various types of pheromones.

7. **Comment:** I've noticed the presence of an overhead camera to track the e-puck positions. Could the authors confirm that this camera is only used to measure the positions in order to quantify the performance (and not for the actual positioning and used by the control algorithm)?

Answer: We clarified that the camera is only used to measure the performance of the swarm.

The overhead camera was used only to measure the performance of the swarm and was not used to provide any information to the robots.

8. **Comment:** I was intrigued by the performance of Overo Gumstick single-board-computer. Was it sufficient to carry out all the onboard simulations?

Answer: We clarified that we run the automatic design processes in a high-performance computational cluster.

The Overo Gusmtick single-board computer is the onboard computer of the robots. The onboard computer executes the control software produced by the design process, and no simulation is carried out on it.

We executed all automatic design processes on a high-performance computational cluster with about 1500 computing cores.

9. **Comment:** The 4 missions considered by the authors are fairly standard. I was wondering if the authors anticipate any issue using Habanero for a more complex mission. What are the limitations?

Answer: We made explicit some important limitations of *Habanero*. These limitations are tightly linked to desired properties for pheromone-based stigmergy that we envision, but we did not consider in the conception of *Habanero*. This point is also discussed in the answer to Comment 6.

It is worth noting that these missions—likewise *Habanero*—are not suitable for drawing conclusions on whether automatic methods can handle more complex missions or design relatively more complex stigmergy-based interactions. For instance, missions that require precise behavioral control via careful modulation of the pheromone deposition and response, or missions that involve more complex communication strategies through various types of pheromones.

REVIEWERS' COMMENTS:

Reviewer #1 (Remarks to the Author):

The authors successfully addressed the issues raised in the comments. The reviewer is happy to recommend to accept the revised manuscript.

Reviewer #2 (Remarks to the Author):

The authors have addressed all of my concerns. I agree that the manuscript is now ready for publication.

Reviewer #3 (Remarks to the Author):

The authors have addressed all my comments and responded satisfactorily to all my questions. I support acceptance and publication of their manuscript.